# In Pursuit of a "Safe" Space for Political Participation: A Study of Selected WhatsApp Communities in Kenya

Gloria Anyango Ooko

Department of Publishing, Journalism and Communication Studies, School of Information Sciences, Moi University, Eldoret 3900-30100, Kenya; glooko15@gmail.com

**Abstract:** Kenya has a history of media censorship and citizen surveillance. The advent of social media is laudable for contributing to freedom of speech and accountability in Kenya. Studies show that WhatsApp, through its group formation affordance, has largely contributed to political participation in Kenya and beyond. Kenyans see it as a "safe" place away from government surveillance, a carry-over of authoritarian rule. This is especially so since WhatsApp is considered as private media compared to other social media platforms. For instance, many political bloggers on Twitter and Facebook perceived to be anti-establishment have been arrested and charged, but only accountable arrests have been made in connection to WhatsApp activities despite government threats. This article argues that although actors, both human and non-human, act to construct a safe community for political participation on WhatsApp, modes of exclusion and inclusion arise from the socio-technological interaction which could pose a threat to the newly founded "safe space". Though the study site is in Kenya, this article grapples with issues other scholars of social media and politics grapple with globally, that is, safety, security, surveillance, and political participation, among others.

**Keywords:** Kenya; social media; WhatsApp; surveillance; privacy; affordances; political participation

## 1. Introduction

The concern for the safety of internet users is persistent in the era of social media and instant messaging apps which encourage individual consumption as opposed to a communal one. The concern is much greater for vulnerable groups in society such as children (Durak 2019; Livingstone et al. 2011, 2017), LGBTQ (Lucero 2017; Scheuerman et al. 2018), women (Clark-Parsons 2018), and the elderly (Zanchetta et al. 2022). Scholars of digital politics are increasingly joining the conversation (Gil de Zúñiga et al. 2021; Pang and Woo 2020; Velasquez et al. 2021). Despite the serendipity surrounding these inventions, instant messaging applications such as WhatsApp have become a popular medium for political mobilization and activism in Africa (Colom 2022; Omanga 2019) and elsewhere (Gil de Zúñiga et al. 2021; Pang and Woo 2020; Treré 2018; Velasquez et al. 2021), especially when they act as tools that enable common people to create online communities.

The instant messaging app WhatsApp, given its "more intimate and controlled environment" (Gil de Zúñiga et al. 2021, p. 201), is particularly popular in Africa which is characterized by several inhibitions related to political participation including media censorship, citizen surveillance, and intrusion of privacy, among others. As such, it provides the much-needed safe environment for political participation often characterized by minority and unpopular views against the establishment, even though some studies report the contrary. For instance, disinformation shared through WhatsApp led to a wave of mob violence and hate killings in India in 2018 (Arun 2019; Farooq 2018). Vasudeva and Barkdull (2020), however, argue that deep-seated fissures within Indian society, coupled with governance failures, only found expression through technology. Consequently, WhatsApp was not responsible for the mob violence, vigilantism, and collapse of the rule of law in India but was just a medium through which these acts were expressed. High rates of

mobile phone ownership in Africa, owing to their increasingly cheaper price (International Telecommunication Union 2020; Wasserman 2011), also contribute to high numbers of WhatsApp use, since the app is mostly accessed on mobile phones as opposed to desktops.

Given the compounded nature of safety on instant messaging apps, this article, therefore, explores ways in which WhatsApp technology affordances enable and/or constrain WhatsApp community members' negotiation of a safe space for maximalist political participation in Kenya. To achieve this, two WhatsApp communities, formed to agitate for political accountability and development from their local governments, were studied. The two are the East Asembo Development Forum (EADF) and Kabula Forward (KF). The communities have been imagined around the notion of place where both East Asembo and Kabula are wards in the Rarieda and Bumula constituencies in Western Kenya.

Based on the main premise of actor–network theory (ANT), the recognition of non-human agency (Latour 2005; Law 1992; Law and Callon 1988), the WhatsApp communities are seen as an assemblage of technology and human actors and, therefore, all these entities' actions are considered in a broader Kenyan political context to answer the following research questions:

1.  How do members' relationships in the actor-network contribute to the construction of a safe space in the two WhatsApp communities?
2.  What strategies are employed by WhatsApp community members to construct a safe space?
3.  To what extent do the strategies work to ensure maximalist participation within the WhatsApp communities?

To answer these research questions, this study employed a netnography method (Kozinets 2007) in which data were collected through background listening (a participant observation strategy), focus group discussions, and in-depth interviews.

## 1.1. Conceptualizing "Safe Space" in Political Discourse

In academia, the conceptualization of a safe space is specific to particular disciplines. For instance, in education discourse, a safe space is generally characterized by an environment where students are free from emotional and psychological harm, their differences are embraced, and they are able to express themselves freely without fear of discrimination (Garcia and Van Soest 1997; Holley and Steiner 2005). In gender studies, safe spaces are seen as those that provide an environment free from violence and discrimination, that allow free expression, especially for marginalized and vulnerable groups such as LGBQTI, women, and the youth (Clark-Parsons 2018; Lucero 2017; Van Heijningen and Van Clief 2017).

In political discourse, conceptualizations of safe spaces mostly follow the critique of the Habermasian public sphere, only that public deliberation has migrated online, a phenomenon dubbed "the digital public sphere." The digital public sphere is considered more inclusive, especially for ordinary citizens, unlike the elitist Habermasian public sphere where a community of intellectual elite led the democratic public discourse (Mahlouly 2013). However, the important argument to be noted about safe spaces is that they do "not necessarily refer to an environment without discomfort, struggle or pain" (Holley and Steiner 2005, p. 50). Online spaces are especially characterized by irrational discourse (Davis 2021; Katiambo 2019) which does not necessarily undermine democracy but rather counters hegemony (see, for example, Bakhtin's (1981) on carnivalesque and Mouffe's (Mouffe 2005) agonistic democracy).

While there is no universal definition of a safe space for political discourse, this article argues that this definition can be inferred from the pillars of democracy including but not limited to freedom (of speech) and inclusivity (Gibson 2019; Dylko and McCluskey 2012) which fosters maximalist participation. After all, Roestone Collective (2014) argues that safe spaces cannot be conceptualized through the static binaries of "safe" and "unsafe" but rather in a relational and contextual manner. As such, the working definition of a safe space in this article is one in which actors (both human and technology), through their action,

negotiate an environment that, to a great extent, fosters egalitarian power relations, a key ingredient for maximalist participation.

### 1.2. The Elusive Safe Space for Political Participation in Kenya

Kenyan citizens have lived under citizen and media surveillance and censorship infringing the democratic space since before independence (Frederiksen 2011; Mbeke 2008), thus making them "unsafe" for political participation. Other issues which lead to mainstream media stifling citizen participation other than surveillance and censorship can be understood under the political-economy analytical lens (see Ogola 2011; Golding and Murdock 1996). Moreover, by design, legacy media are constrained in enabling participatory practices. In this article, the notion of space, understood in hybrid terms as socially constructed while acknowledging its materiality (Latour 1998, cited in Schroer 2019), is placed stark in the middle of the political participation debate in and through the media (Carpentier 2007). For instance, from the colonial period to the beginning of the 21st century, the mainstream media had been located centrally in Nairobi, Kenya's capital city.

Given the then-poor infrastructure of the country in terms of communication and road networks, it meant ordinary citizens in rural areas, the most populous, were excluded from media access. Here, access is used broadly not just in terms of obtaining media content and interpretations but participating in content production and media institutions (Carpentier 2012). To be located elsewhere other than Nairobi, the capital city and center of power, meant to be excluded from the arena where all sectors of society play out. Other centers of influence, counties, however, have since emerged through the decentralization of power, resources, and decision-making following the promulgation of the Constitution of Kenya in 2010.

Another example of how spaces have advanced exclusion and, consequently, inhibited the political participation of the ordinary citizen in Kenya is in the construction of a dichotomy between the "opposition" and "government supporters". The administrative partitioning of Kenya maintains the colonial design since the units are divided along tribal or communal fault lines. Successive post-colonial authoritarian regimes have continued this colonial legacy. Ethnic groups are coterminous, with administrative boundaries that enable politicians to canvass for support along ethnic lines which contributes to ethnic voting patterns in Kenya. A government versus opposition supporters dichotomy tends to arise but fizzles out owing to the dominance of fickle personality cult politics impervious to programmatic policies and actions. National resources and infrastructure development, which impact legacy media location and access, tend to be allocated unevenly in favor of government supporters (Shilaho 2018).

It is, therefore, illuminating that the same strategy of the occupation of space, exploited by oppressors, has also been appropriated for the safety of political mobilization by citizens to counter oppression. "Jeevanjee Gardens", "Kamukunji", and "Freedom Corner" in Nairobi are historical incubation spaces for political activism in Kenya. These spaces are crucibles of activism by civil society, filling in the gap left by the mainstream media who have failed in their watchdog role, due to self-censorship, government control, and co-option by the government since some of these media companies are owned either by government officials, including the immediate former president, Uhuru Kenyatta, or individuals connected to successive governments in Kenya.

However, activism spaces have been infiltrated by government informers who thwart and compromise mobilization efforts. Brute force by the police is used to achieve the same end. Initially, Bunge la Mwananchi or "People's Parliament", an organic social movement in Kenya based in Jeevanjee Gardens, was hailed by Kimari and Rasmussen (2010) for its resistance to institutionalization and dominance by political actors unlike other social movements that are registered by non-governmental organizations and susceptible to ideological meddling by donors (Willems 2015). Bunge la Mwananchi, however, seems to have been compromised too. For instance, what Kimari and Rasmussen (2010) described as the "choosing of a ceremonial leader" by Bunge la Mwananchi members, turned into

fully fledged elections with polarizing campaigns, which created room for politicians to influence the results by funding the campaigns.

There are other challenges that have weakened civil society in Kenya and, indeed, across Africa including a hostile political environment, a lack of funding, corruption, authoritarian tendencies, and co-optation into government (Munene and Thakhathi 2017; Wanyande 2009). Availability of funding could also be said to render these organizations inherently weak because it robs them of strategic independence and ideational autonomy. The association of the offline "safe" spaces with compromised civil society organizations has, therefore, rendered them ineffective, creating a vacuum and the desire for new "safe" spaces for counter-hegemonic discourses. This, coupled with the inability of the legacy media to offer audiences more participatory spaces, has led audiences in Africa to move to digital media (Willems and Mano 2017). One such space in Kenya, which is the focus of this article, is the WhatsApp community enabled by the group formation affordance of the technology.

Just as is the case with the legacy media, other internet-based media including social media platforms such as Facebook and Twitter, which preceded WhatsApp, have also contributed to the view of WhatsApp providing a safe space for political agitation. According to Mukhongo (2020), the blog facilitated one of the earliest forms of collective interaction on the internet in Kenya. She cites the Kenya Bloggers Webring (KBW), launched in 2003 by Daudi Were and Mashada.com, as pioneer blogs which provided both local and international information to Kenyans (Mukhongo 2020). The personal blog, "Kenyan Pundit" by Ory Okolloh, one of the founders of Ushahidi, offers commentary, particularly political commentary which invites her audience to critically consider political deficiencies in Sub-Saharan Africa (Soetan 2012).

Ushahidi (Swahili for testimony), an open-source software application which allows for data crowdsourcing and analysis, in its advent during the 2007/2008 post-election violence in Kenya, provided a counter-public sphere through which citizens could send messages which helped in mapping the areas affected by violence. It provided the necessary safe space for ordinary citizens to be heard and warned of potential danger as the government had banned mainstream media live coverage under the guise of it fueling violence, at a time when the watchdog and surveillance roles of the media were most crucial. It has since been shut down. The Ushahidi platform has also been instrumental in the expansion of democracy in other countries in Africa including Nigeria. For instance, during the 2011 national election in Nigeria, the collective movement ReclaimNaija used the Ushahidi application for crowdsourcing election monitoring (Bailard and Livingston 2014). By doing so, they provided an important public good, that is, information on the election processes, especially where there was system failure, documenting, archiving, and making it accessible for the government and system at large. It is noteworthy that the Ushahidi platform managed to provide this good whereas government and other civil society organizations lacked the capacity to provide the same (Bailard and Livingston 2014). Of importance is that these platforms also allow for citizen-led initiatives, in this case, domestic election monitoring which builds more trust in the election process as opposed to international-led election monitoring which is often looked at with suspicion amid the sovereignty interference question (Sassetti 2019).

Another such platform, *Uchaguzi*, Kiswahili for decision, followed in 2010, just prior to the referendum that led to the promulgation of the current Kenyan Constitution. It uses the mobile phone platform through which citizens can report electoral malpractices and other experiences (Aarvik 2015).

There are various studies on social media which not only demonstrate the idea of a free and safe space but also show how online communities can be organized intentionally or inadvertently along antagonistic lines of us vs them. These communities also provide alternative spaces to the mainstream media for collective action. Kenyans on Twitter (KOT) is a prominent example of these communities, with Mutahi and Kimari (2017) observing that the hashtag #KOT is often used to mobilize Kenyans on Twitter around social action

and political accountability. KOT comprises regular Kenyan users of Twitter who use the hashtag #KOT to engage in the everyday happenings in the country. Part of that everyday touches on political matters and on such occasions, the hashtag #KOT serves as a collective call through which Kenyans are mobilized on social media to participate in political activism. This was the case with the campaign #SomeoneTellCNN used by KOT to launch an uncivil protest against CNN in the wake of Obama's visit to Kenya (Katiambo 2017).

Additionally, Okoth (2020) shows how KOT frequently use visuals including memes, newspaper and Google screenshots to poke fun and ridicule their leaders in the bid of holding them accountable. Diepeveen (2019), using a Facebook group named "Mombasa Youth Senate", explores the political nature of discussions in this group and finds that Facebook structures and users' experience do not encourage new imaginaries on citizen–state relations. Regardless, Diepeveen acknowledges the open space for interaction that Facebook provides its users.

In a study of a WhatsApp group, Katiambo and Ochoti (2021) explain how a seemingly meaningless talk on predatory publishing with the boundaries being between "fake and peer reviewed" journals on a WhatsApp group of lecturers in Kenya, "reveal struggles at the unconscious level in the era of corporatized universities and alternative facts" (p. 39). Additionally, Omanga (2019) discusses how discourses on WhatsApp do not only mobilize participants toward collective political action online, but also lead to offline activism.

The explosion of digital media and other information communication technologies, particularly mobile technology, can, therefore, be said to be contributing, to a great extent, to democracy and its derivatives including political stability and good governance (Bailard and Livingston 2014; Livingston 2011; Livingston and Walter-Drop 2014; Sassetti 2019).

### 1.3. Presenting Maximalist Participation as an Ingredient for a Political "Safe" Space

As stated before, this article argues that a safe space for political participation is one in which actors (both human and technology), through their action, negotiate an environment that, to a great extent, fosters egalitarian power relations, a key ingredient for maximalist participation.

Wenger (1998, p. 55) defines participation "as the social experience of living in the world in terms of membership in social communities and active involvement in social enterprises". Others see participation as a political process in which power struggles are inherent (Carpentier 2007, 2014; Laclau and Mouffe 1985). For instance, Carpentier (2014) defines participation as a "situation where the actors involved in (formal or informal) decision making processes are positioned towards each other through power relationships that are (to some extent) egalitarian" (p. 1002). Moreover, according to Carpentier (2009), debates on participation should analyze the distribution of power within the society both at macro and micro levels. Participation, therefore, implies a situation in which a power imbalance naturally exists and individuals, through the same process of participation, try to equalize it (Carpentier et al. 2013).

In community media discourse, participation takes a dichotomy of participation through (minimalist) and in the media (maximalist) (Carpentier et al. 2003) where in the former, the media simply act as a platform for citizens to air their views and interact with others while the latter goes beyond simple access and interaction to provide an opportunity for ordinary citizens to contribute in both the generation of media content and decision-making (Carpentier et al. 2003). The latter is associated with a more democratic media which is the ideal that community media should strive to achieve (Carpentier et al. 2013; Carpentier 2012).

In studies of the organization of activism on social media, Bruns et al. (2008) and Shirky (2009) posit that the internet allows for a more equal distribution of power among members of a community through deeper participation. However, this is not usually the case, as Svensson (2012), following Foucault (1979, 1994) and Elias (1970), argues that since participation is a process that takes place among people, it is unlikely that participation

hierarchies and power relationships can be eliminated. The internet has previously been theorized as "a safe place" where individuals can speak their mind freely, a necessary ingredient for maximalist participation that studies have shown is hard to achieve (Katzer et al. 2009; Pearce and Vital 2015; Rheingold 2000). Matei (2005) argues that anonymity in internet-based communities, achieved by using pseudo accounts, fosters an increase in participation.

However, through power sideshows including hostile hazing rituals, individuals in online communities exclude some members from participation (Seabrook 1998). In the WhatsApp communities studied, exclusion happens when members deemed to be going against the groups' purpose are removed by the administrator who can either make that decision solely or upon request by other members.

Moreover, the materiality of technology further rearticulates the nature of participation in virtual spaces. For instance, the presence of "administrators" (individuals who form WhatsApp groups and can admit or remove members) in WhatsApp and other virtual communities implies moderation of members' behavior. Social media ownership also feeds discourse on the capitalistic exploitation of digital subjectivities, especially by social networking platforms (Fisher 2009; Fuchs 2011; Grosser 2014), while giving the illusion of encouraging political agency (Dean 2012) and participation (Andrejevic 2013; Lovink 2011). Since online communities, like older formulations of community, manifest strife and struggle, given the caution from various scholars not to confuse safe spaces as free from contestations, can WhatsApp communities be constituted as safe for political participation?

### 1.4. Free Speech, Surveillance, and "Safe Spaces"

Free speech is the crux of democracy in which dominant political ideologies can be countered and political leaders held accountable without censorship or restraint. While since their inception, online platforms have been hailed for expanding opportunities for freedom of expression and democracy through increased access to information and participation (Jenkins 2006; Rheingold 2000), they have also been critiqued for restraining it. In Kenya, for instance, Omanga (2019) explains how a chief has transformed the traditional *baraza* (local public meeting) into a more effective online *baraza* on Twitter. In another study, Ogola (2015) found that key bloggers/activists and citizen journalists on Twitter have created a more horizontal participatory environment on wider national narratives amid the platform's own hierarchies. Flichy (2010) cited in Mahlouly (2013) observes that the digital public sphere is inclusive and diverse in terms of the nature of participants, especially when juxtaposed against the Habermasian public sphere. In the same breath, however, he casts a shadow on the quality of online participation, citing the poor quality of political debate that results from this plurality while also noting the unsustainability of online political engagements as he argues that they are mostly ad hoc depending on individual interests.

More radically, Papacharissi (2009, p. 234) argues that "while the internet and surrounding digital technologies provide a public space, they do not necessarily provide a public sphere." A public space moves from the public sphere conceptualization of rational participants to acknowledge the need for "passion in politics" and, therefore, the possibility of agonism (Mouffe, in an interview with Carpentier and Cammaerts (2006, p. 11)). Moreover, studies on social media in Africa have described publics as "unruly" (Srinivasan et al. 2019, p. 55) and the digital space or realm as "emerging as fragmented, transient, polarizing and unreliable" (Srinivasan et al. 2019, p. 3).

Surveillance and invasion of privacy also contribute to the questioning of the construction of WhatsApp as a safe space. Studies on internet surveillance consider various aspects including vertical surveillance (Foucault 1985), inverted forms such as sousveillance (Mann et al. 2003), and more socially interactive surveillance (Albrechtslund 2008). Furthermore, those under watch are not always passive subjects but may welcome the surveillance because of certain benefits they gain from it. Of importance to note is the

liquidity of the concept of surveillance (Bauman and Lyon 2012) including its dual working of both care and control (Lyon 1994).

Kenya has enacted several laws including the Computer Misuse and Cybercrimes Act, 2018 (Republic of Kenya 2018), which aims to criminalize the resistant activities of citizens in the guise of "misuse" of the internet. On the other hand, WhatsApp has a security feature known as end-to-end encryption (E2EE) which, according to the app owners, means that "data exchanged between two communicating parties is encrypted in a way that only the sender and the intended recipient can decrypt it, so, e.g., eavesdroppers and service providers cannot read or modify messages" (Onwuzurike and Cristofaro 2016, p. 2).

Based on these laws, mobile phone users are monitored through the IMEI3 codes as it is a requirement that they must register their mobile numbers using their national IDs, and failure to do so will lead to their SIM cards being blocked. The International Mobile Equipment Identity or IMEI is a number, usually unique, to identify 3rd Generation Partnership Project (3GPP) and Integrated Digital Enhanced Network (iDEN) mobile phones, as well as some satellite phones (Jha and Krasner 2008).

In July 2017, just before Kenya's general election, the Chairman of the National Cohesion and Integration Commission (NCIC) announced that they were investigating hate-speech mongers on 21 WhatsApp groups and warned that the administrators of groups involved in hate speech would be arrested. Consequently, a few WhatsApp administrators were arrested and charged in relation to their WhatsApp activities. Those implicated in spreading hateful and inciteful messages on Facebook were charged too, which suggested the shareability affordance of WhatsApp technology might not be tamperproof. The E2EE may not be as foolproof as claimed by WhatsApp's owners, Facebook.

Another issue related to surveillance and participation is privacy. Earlier, scholars such as Martin Heiddeger presented the private sphere as the only space where one can be themselves and express themselves freely. However, ICTs have blurred the difference between what is "private" and "public". Leary and Kowalski (1995), for instance, argue that "private" and "public" is not a matter of either/or but rather the chance that "one's behavior will be observed by others and the number of others who might see or learn about it" (p. 26). Boyd (2010, p. 45) gives a more technological affordance view on this. She argues that affordances of networked sites, such as persistence, replicability, scalability, and searchability, determine through the social behaviors they cultivate, how one moves back and forth from private to public. For instance, the shareability affordance in WhatsApp means that individuals can take screen shots of messages and spread them to other WhatsApp users and even across platforms such as Facebook, Twitter, and e-mail, rendering these messages public. The fear of what is privately shared being made public can deter one from participation in digital spaces.

## 2. Theoretical Concepts: Actor–Network Theory (ANT)

This article will not give an account of the entire theory but will only focus on the ontology of the social, non-human actor agency, and performativity as is relevant to the aim of this study. ANT is summarized by Law (2009, p. 141) as "a disparate family of material-semiotic tools, sensibilities, and methods of analysis that treat everything in the social and natural worlds as a continuously generated effect of the webs of relations within which they are located." This definition precipitates the argument and methodology of this study that the material and the discursive are interconnected, social media safe spaces are constructed and negotiated by actors' "doings" and, thus, it is not a space of stability but rather a fluid space characterized by a back-and-forth movement between "safe" and "unsafe".

Latour (2005) is against the a priori stability granted to the social. He instead defines the social as "a trail of associations between heterogeneous elements" (Latour 2005, p. 5) making sociology "the tracing of associations" (Latour 2005, p. 5). What Latour (2005) means is that the main focus of social inquiry should be how the social is produced, that is, how actor-networks are formed and how they might collapse instead of the "why" the social is as it is. It is fluid rather than static. Tracing this process of production involves

"following the actors" (Latour 2005, p. 12) as they associate, assemble, and reassemble as "heterogeneous elements might be assembled a new in some given state of affairs" (Latour 2005, p. 5). An actor-network, therefore, refers to a group of heterogeneous elements interconnected and affecting each other through different relationships (Law 1992). It is these interconnections and associations that ANT studies focus on. This is not to say that the primary focus of ANT is interactions among individuals in a network but how actors "define and distribute roles and mobilize or invent others to play these roles" (Law and Callon 1988, p. 285). The actor-network under study in this case is the assemblage, to use Deleuze and Guattari's (1987) parlance, of WhatsApp technology, members of the virtual communities enabled by WhatsApp, and the external environment such as the culture and politics within which this network operates.

The second important tenet of ANT this article draws from is that it accords both human and non-human actors' agency to act. As such, an actor is a "semiotic definition-an actant-that is something that acts or to which activity is granted by others (Latour 1996, p. 373); literally anything that exert detectable influence on others" (Law 1987, p. 132). Ability to act is not "a property of humans but of an association of actant" (Latour 1999, p. 182). Consequently, how the WhatsApp technology acts through its features and affordances is equally considered alongside the actions of human members of these WhatsApp groups while understanding how their "collective action" constructs a safe community for political participation or not.

Thirdly, this article draws from ANT, the concept of performativity. Latour's view of performativity develops from his theorization of power in "practice" which results from the collective action of actors. That is, "the intense activity of enrolling, convincing and enlisting" (Latour 2005, p. 273), what is referred to as translation in ANT. In this formulation, action goes beyond Austin's speech acts to the practices that create and maintain networks (Ringmar 2018). The reiterability of these actions is important as they provide a pattern through which actors can be followed and give meaning to the actors' practices.

## 3. Materials and Methods

This section gives an account of the philosophical underpinnings and the specific research methods employed in this study.

### 3.1. The Cases

Two instrumental cases, East Asembo Development Forum (EADF) and Kabula Forward (KF) WhatsApp communities, were selected purposefully as they fit within the criteria of being formed for political interest, that of holding political leaders accountable and agitating for development. They are typical of other groups formed to facilitate political participation on Facebook and other social media platforms. However, they are also unique in terms of the platform, that is, WhatsApp, rendering them more private compared to such groups on more public platforms. Purposive sampling is the selection of a sample based on a particular purpose (Marshall 1996) and information expectations on the content the researcher is interested in (Flyvbjerg 2001). The communities are imagined spatially through a sense of place. East Asembo and Kabula are wards in Western Kenya predominantly occupied by the Luo and Bukusu people, respectively. Wards are the smallest forms of administration for county governments, the second tier of government. In both WhatsApp communities, membership comprises individuals residing in both rural and urban areas (including the diaspora). At the time of data collection, while KF had 230 members, EADF had 182 members, though the numbers kept changing as people left, were "removed", or added into the groups. At the time of this study, a WhatsApp group could only hold a maximum number of 256 members. Moreover, data were collected post-general election to capture the everyday discourses and avoid the bias that an election period could bring forth in terms of numbers and constant heated debates. This explains the somewhat "low" numbers within the WhatsApp communities. A WhatsApp group limit has since been increased by Meta to 1024 concurrent participants. In both communities, there were fewer

female than male members. Furthermore, the majority of members of EADF fell within the youth age bracket (between 18–35) while for KF, most members were above 35.

The case study approach has been faulted for being time-consuming, costly, prone to bias, and a minimal basis for scientific generalization. Given the small number of participants/cases researched (what is considered its major weakness), this study employed instrumental case studies which are seen to overcome these disadvantages (Stake 1995). For instance, while Mintzberg et al. (2005, p. 10) argue that "If there is no generalizing beyond the data, no theory. No theory, no insight. And if no insight, why do research?" instrumental case studies overcome the need for generalization as the cases are a means of understanding something more than just the particular cases. Consequently, EADF and KF play a supportive role in the achievement of the aim of this study, that is, understanding how WhatsApp communities are constructed as safe spaces. As Stake (1995) argues, the focus of case studies is not statistical generalization but rather in making sense of a particular, the problematics of safety in WhatsApp communities. The EADF and KF cases are, however, looked at in depth and their ordinary activities detailed in terms of their typicality to other cases or their uniqueness because doing so helps in pursuing the goal of this study (Stake 1995; Yin 2009).

### 3.2. Methods

This study follows an interpretivist ontology and constructivism epistemology in which reality and ways of knowing are seen as multiple. As such, both the human and material components (WhatsApp technology) contribute to the discourses of a safe space for political participation, informing the multimodal data collection. Further ethnographic methods, in particular netnography, were employed. Put forward by Kozinets (2007), netnography is a form of ethnography where, as in traditional ethnography, the major principle employed is immersion into the community of study. Netnography, which follows the same conceptualization as "virtual ethnography" (Hine 2008), "digital ethnography" (Murthy 2008), and social media ethnography (Postill and Pink 2012), also applies a multi-sited field work with the intermittent employment of online participant observations and offline interviews and focus group discussions (FGDs). Mare (2017), who used social media ethnography to explain the political action of youth on Facebook in Zimbabwe and South Africa, and Omanga (2019), who employed digital ethnography to explore how WhatsApp consists of "digital publics" in Kenya, illustrate the robustness of these methods. Like physical spaces, virtual spaces have a particular language and style of engagement determined by technological affordances which also require the researcher to spend time in the community to understand what occurs there (Mare 2017).

Data were collected through background listening (metaphoric for voice) which is a proactive and more participatory method that overcomes the shortcomings and stigma surrounding lurking (Crawford 2009). The method allows one to be a participant observer, following without influencing the WhatsApp communities' discussions. Semi-structured interviews, mostly conducted offline, were also used to obtain in-depth information and triangulate data from observations. In the case of translocal members, online interviews via WhatsApp were conducted. A purposeful selection of interview samples of active, least active, those who had exited the communities, and those who are not locally based in the places in question was collated to obtain various viewpoints.

The sample consisted of 2 main administrators of the WhatsApp communities, 10 local, 10 translocal members, and 5 members who exited the communities (when one exits, this information is automatically displayed on the WhatsApp community page, an affordance of WhatsApp technology).

WhatsApp texts (posted online between August 2017 and December 2018) were first mined randomly using a data mining application by entering certain keywords—power, participation, active, government, comment, rules, decision, and post—into it. The keywords were derived from the purpose of the article and a review of the literature on online groups. In the second phase, the data that fell within discursive moments

were purposefully selected, guiding the selection of interview participants. A discursive moment refers to a critical juncture in discourse (Reed 2005) which disrupts the "normal" trajectory of discourse in a social setup. In this study, these junctures were identified as moments where a tangible action such as decision-making was required of members in the WhatsApp communities.

Discourse material analysis (DMA) was used to analyze collected data, combined with a reading of the data through the actor–network theory (ANT) lens, focusing on the techniques actors use to perform their agency in the construction of WhatsApp as a safe space. Specifically, analysis is guided by three tenets of ANT: the ontology of the social against the a priori stability of the social (Latour 2005), agency to all actors in a network including the material elements, and lastly, performativity which analyzes the collective action of actors that create and maintain networks (Ringmar 2018).

### 3.3. Procedure for Data Analysis

In line with ANT's ontological premise, which accords both human and non-human actants agency to act, discursive material analysis (DMA), in which both materiality and symbolic practices of the WhatsApp assemblages are read as discursive practices, was employed for analysis. Furthermore, as Bennett (2004, p. 348) articulates, technological materialism not only aids social practices but also shapes how people experience the social. These practices are understood in the sense of performative actions as provided by ANT. Specifically, and in line with a non-hierarchical treatment of the discursive and material agency, what I call a techno-trope analysis was applied to enable concurrent analysis. The neologism "techno-trope" combines the technology aspect of WhatsApp and rhetorical tropes. The idea that the naming of technological affordances and features, though may be read literally, also carry figurative functions that are used politically in modes of inclusion and exclusion in WhatsApp communities. Rhetoric tropes, at the ontic level, refer to the symbolic use of words to invoke meanings, different from the literal meaning of those words. They include, for instance, metonyms, metaphors, and catachresis. However, when used at an ontological level of discourse, rhetoric goes beyond the use of figurative language as opposed to literal meaning, to take a political meaning, "that specifies the interweaving of words and actions into practices, the contingency of all identity, the primacy of politics, and so forth" (Howarth and Griggs 2006, p. 29). In this article, metaphors at the ontic and ontological level were used to analyze the practices of the WhatsApp assemblage through its affordances (including agency, modality, navigability, and interactivity) and how they inform technology and human actors' actions toward negotiating safety in these spaces.

Data from interviews and FGDs were first transcribed, collated, edited, and coded into emerging themes in relation to the research questions and the stipulated affordances. The discussions (observation data) on WhatsApp covering the stipulated period were first exported to Excel through the WhatsApp "export" feature which also organizes the information into meaningful categories of "date, time, speaker, and what is spoken" and further organized thematically. A search through the organized data using keywords and phrases derived from the relevant literature review and the data collection was used to obtain content relevant to the research questions. The keywords include "add, remove, left, join, fear/afraid, government, rules, laws, safe, punish, community", among others. The content was then interpreted and discussed according to the metaphoric practices they present. For instance, the "add" feature of WhatsApp is not only understood literally as enabling actors to join a WhatsApp community assemblage but also metaphorically as an agential political practice of inclusion and exclusion, contributing consecutively to "safe" and "unsafe" WhatsApp spaces.

### 3.4. Ethical Considerations

Conducting online research comes with unique ethical and even legal challenges in some cases. Given that when applying traditional ethical guidelines, online research can

be problematic (Grinyer 2007), the researcher followed Kozinets' (2002) call for one to be "flexible" and "unobtrusive" (p. 70).

Tuikka et al. (2017), having analyzed 52 articles which use netnography, adapts 3 of Kozinets' (2002, 2010) major ethical questions. They argue that netnographic research can be conducted in many different ethically justified ways provided the researcher considers the following questions:

(a) Do you need to ask the informed consent of the members of the online community in question?

(b) Do you need to protect the anonymity of the members of the online community in question?

(c) How important is the accountability of your research? (Tuikka et al. 2017, p. 9.)

In this study, the answers to all the above questions were in the positive. To begin with, to use WhatsApp, a user needs to register on the platform with their phone number to create an account. To join a WhatsApp community, they need to be added by an administrator or join using a provided group link. Since the WhatsApp communities under study are restricted to those who come from a particular place, access to these communities is dependent on this fact and the rules and purpose of the communities, holding their leaders accountable and pushing for development abound. Since I was joining them for the purpose of research, I needed to obtain informed consent from members. Given the nature of these communities, the precarious laws guiding usage of social media in Kenya, and arrest of some WhatsApp administrators of other communities, there was a need to maintain anonymity. The members of these communities laid down their expectations during an open session I held with them which touched on confidentiality and privacy. I am, therefore, accountable for my decisions as the researcher. As such, all the guidelines put forward by Kozinets (2002) are useful in this study.

I applied the first guideline by letting members know, starting with the administrators, that I was part of the communities for research. This was achieved through a post in the communities, stating who I was, my purpose, and sought consent from members to observe what transpired in the communities. As is the critique of Kozinets' (2002) full researcher disclosure ethical component, which Posey et al. (2010) term as restrictive, I encountered opposition from some members and communities. Kozinets (2015) and Xun and Reynolds (2010) also recognize the risk disclosure may pose to research as not all participants may consent to being "watched" by a researcher. Roy et al. (2015) advise that a researcher can minimize the risk of losing data, especially for private groups as is the case of WhatsApp communities, by contacting as many as ten groups until they find one that would allow researcher participation.

I started out with four communities and ended up losing two that were not comfortable in participating in the research. For the two, EADF and KF, that gave informed consent, I had an open session with them within the communities where I allowed them to ask questions and suggest ground rules on researcher's conduct. The session proved productive as members informed me of what they expected, for instance, "no screenshots," "anonymity," though in some cases members told me directly that they did not mind being named. Announcing my presence early enough allowed me and the communities to go through the motions from suspicion to trust and to cover instances of membership changes given the fluidity of online communities (Sugiura et al. 2017). Early disclosure (in 2016) also meant that I could fall into the background where participants "forgot" about my existence and interacted without my presence affecting them. Despite falling in the background, I was, however, aware at all times of the power my positionality as a researcher in the two WhatsApp communities accorded me. It is for this reason that I only included a much later data set of postings between August 2017 and December 2018 to minimize chances that I could have influenced interaction in any way. Moreover, allowing a discussion on what the community members' expectations of how I should conduct myself during the research process also helped me to a certain extent to give back power to the research participants.

To minimize breach of confidentiality and intrusion of privacy caused by traceability, I removed personal identifiers from the chat log and used pseudonyms instead. I also avoided taking screenshots of postings in the WhatsApp communities. Feedback was obtained, especially during individual interviews where I would show interviewees their conversations on the WhatsApp community platform and ask them to clarify what they meant and whether they still feel the same way. The fourth component needed creativity to navigate, especially since the line between private and public in online spaces keeps blurring. The same could be argued for WhatsApp which, though considered private, as one would not know of a community's existence unless they are added to it or invited via a link and is also equipped with the end-to-end encryption privacy feature, a challenge to this privacy is possible through what the technology itself allows. For instance, taking and sharing screenshots across other more public platforms.

In the case of this study, participants insisted that I should not use screenshots of their online postings and that request was upheld, even though some participants opined that their activities on WhatsApp do not define who they are or affect their offline personalities. Additionally, the chat logs data exported to my email were solely used for research and not shared with anyone else. Screenshots could be likened to direct quotes which Skågeby (2011) argues should not be used in the research report as they compromise anonymity, especially if that information could possibly be harmful to the participants (Tuikka et al. 2017).

## 4. Results

This section is a discussion of the findings of this study, organized chronologically to address the research questions. The first finding shows how technological affordances enable exclusionary practices and the building of a political frontier through the construction of a discursive relationship of "us" versus "them" which defines how members understand a safe space. The second finding outlines the strategies members use to negotiate safety in WhatsApp while the third examines the extent to which socio-technological affordances and members' strategies lead to maximalist participation, the key definer of a safe space in the context of the WhatsApp communities under study.

### 4.1. Mapping Relationships in an Online Community

This finding maps the relationship among the actors in the WhatsApp communities by "following" them as they enroll, mobilize into different roles and relationships, and negotiate a safe space within the networks (Law and Callon 1988).

4.1.1. Enrolment into the WhatsApp Communities

Enrolment into the network is a combined affordance of both the technological and human actors, initiated by the technology actor through the "add" feature of a WhatsApp group or the "invite" link. These features go beyond the literal means through which members can join the communities, but also act as metaphorical modes of inclusion and exclusion. Discursively, the individual who starts the group/community becomes the technology-designated leader or an administrator by WhatsApp terms who can then add members into the group using their mobile phone numbers. However, enrolment is not performed solely by the administrators but other members also recommend friends to be added in the group. It is at this stage that "threats" of exclusion would begin. For instance, when asked how they decide who to add in the group, one of the administrators responded:

> "When I formed the group, I added people I knew, who I thought would contribute meaningfully towards the goal of the group. It is hard to add somebody you do not know because you do not have their mobile phone numbers. Those added also usually suggest people to be added by forwarding their numbers to me". (Administrator, interview, EADF, 2018)

The administrators also indicated that they did not use the option of the invite link to recruit members as they wanted to limit it to people who come from East Asembo and Kabula, respectively, as with the link, anybody who accesses it can join the group with just

a click. This points to how the relationship of belonging in the actor-network by virtue of hailing from either administrative ward is maintained.

### 4.1.2. Composition of EADF and KF WhatsApp Communities

The actor-network assemblages of the WhatsApp communities point to heterogenous actors; ordinary citizens (both locally based and, in the diaspora (translocal)), the WhatsApp technology, and government through membership and the policies it makes with regard to social media use. Government is used broadly to include the local political leaders and individuals in the public service such as policemen and chiefs. To the question regarding the goal of the communities and why they joined, the following are some of the participants' responses:

> "I formed KF on the WhatsApp platform because I saw the need to bring us 'common wananchi (citizens)' together to ensure that our leaders do the work they were elected to do and to find ways through which our community can develop". (Super administrator, interview, KF 2018)

> "EADF was formed because e did not have a space as common members of East Asembo ward to freely discuss development issues affecting us. Our leaders like the Member of County Assembly (MCA), the Member of Parliament – MP and even the chief are here. We can therefore approach them directly with our issues and demand they act accordingly because we elected them to serve us. They cannot hide away from us as they used to before we had this platform". (Kilian, interview, EADF 2018)

> "I joined EADF because even though I live in the UK, East Asembo is my home, I was born there, my family lives there. And therefore, as a citizen of that place, I need to contribute to its development and that means following closely what the elected leaders do, especially our Member of the County Assembly (MCA). It is not proper to leave such matters to only those who are based at home" (refers to rural dwellers locally based in their respective ward). (Phoebe, interview, EADF 2018)

The goal of the WhatsApp communities and co-existence of ordinary citizens with political leaders in the same space create a political frontier which positions the ordinary citizen (we) against a constructed enemy (them), that is, the politicians. The unity of purpose among ordinary citizens contributes to feelings of safety which allow free speech, an important ingredient for political participation. Political leaders, therefore, become the other in these online spaces, contrary to offline places, as exemplified in the excerpts below in response to how participants feel about the political leaders being part of their WhatsApp community:

> "We welcome the presence of our political leaders; we do not fear them at all nor do they influence what we say in the platform. In fact, we talk about their shortcomings and lack of focus on development because we know they are in the group and will get to know how we feel about them". (Kerry, FGD, EADF 2018)

> "When I heard about EADF, I told myself, I must be part of it. You know when you are a leader, you are always looking for an opportunity to hear first-hand what the people you represent are saying about it. I use what is said about my leadership in EADF to improve. Even though sometimes people say very hurtful things and you are tempted to quit, but I stay put. I have since decided to keep quiet and not argue with everyone on the platform, I only speak when asked directly to address an issue". (elected Member of County Assembly (MCA), interview, EADF 2018)

### 4.1.3. Rules Governing Interaction

Despite the heterogeneity of actors in the network, their combined agency and, consequently, acts work together to create a whole, the WhatsApp network. One of the actions

that glues associations in the networks is that there is a definite modus operandi to their interactions. From participant observations, it emerged that both communities do not have written rules of engagement. However, culture largely informs how members relate to one another. In both communities, for instance, the elderly have to be respected. This then means that younger members cannot act uncivilly toward older members of the virtual communities even though they may disagree with their opinion, creating a friendly environment for participation. This does not mean there are no instances where acts considered "uncivil" by members are performed. When this happens, usually the "errant" member is warned, asked to apologize, or excluded from the virtual community by being removed if the action is repeated. The iteration in the way interaction is governed further strengthens the actor-network (Ringmar 2018). The following excerpts show some responses to the question of whether there were rules governing conduct and how actors formed them:

> "KF does not have any written rules. We also did not discuss any rules of engagement, but naturally when you interact with people, there are certain things that are obvious. For instance, you have to respect your elders, you do not talk back or argue unnecessarily, it is our culture. You also need to give people a chance to say what they want to say, you do not insult them unless you are joking with your friends and age mates". (Lori, interview, KF 2018)

This sentiment is also shared by members of EADF as demonstrated by Onyango (interview, 2018):

> "Since I joined this group, we have never discussed any rules, but some things are obvious when people come together. We are not supposed to take screen shots for instance or share conversations here elsewhere. Respect is important. Our culture informs how we engage."

What happens when one is deemed not to adhere to the agreed rules is what contributes to the construction of a safe space. WhatsApp technology allows the administrator to remove any member of the group if they want to:

> "If you are cautioned to stop engaging in things that go against the agreed rules and you do not stop, then the admin can remove you from the group. Sometimes, members demand that you be removed. If this happens, you will miss what is happening in Kabula Forward. So many things are discussed on this forum. You really feel bad if you are not here". (Shifefwe, FGD, KF 2018)

*4.2. Negotiating Safety: "What happens in WhatsApp Remains in WhatsApp!" or Does It?*

This finding speaks to discursive strategies derived from interviews, FGDs, and observations that participants in the WhatsApp communities employ to negotiate safety, despite noting that certain affordances meant to ensure security of information shared in the platform are, in fact, false affordances.

4.2.1. Trust in WhatsApp Technology Affordances

From participant observations, confirmed during interviews, the desire to belong, to have a platform on which one's voice can be heard, supersedes the fear of surveillance or breach of privacy. Moreover, some participants cited their trust in some of the WhatsApp affordances as an assurance of safety from surveillance, consequently influencing their active participation. For instance, regarding the question of "Why WhatsApp?" a participant indicated:

> "We decided to have EADF on WhatsApp because unlike other social media like Facebook and twitter, A WhatsApp group is private. It is hard to tell it exists which reduces the likelihood that the government will spy on you. We are free and safe to discuss political issues without fear". (Pat, interview, EADF 2018)

Another technology safety affordance welcomed by participants is the end-to-end encryption (E2EE) meant to prevent third parties from accessing information from the WhatsApp platform. The E2EE metaphorically contributes positively to interaction as it gives partici-

pants a sense of safety. Participants are nevertheless not naïve to think that these security features in WhatsApp afford them total safety. If anything, they acknowledged that apart from the fact that people can take screenshots of conversations and share them on other public platforms, the government is capable of hacking WhatsApp if need be. However, they were not worried because "what happens in WhatsApp remains in WhatsApp", quipped a participant:

> "I understand that director of the National Cohesion and Integration Commission issued warning that administrators will be arrested if members of a WhatsApp group engage in hate speech, but so far, no administrator has been arrested. I have heard that some people who insulted the president on FB (Facebook) were arrested and jailed, but so far no arrest on WhatsApp users. Don't you think if the government 'knew' what was happening in WhatsApp, they would have already arrested some people? Sometimes we quarrel on WhatsApp and say wrong things but the admin always cautions us on these matters, other members can also caution you. One time, I met a member who had insulted me on WhatsApp, when I asked him about it, he said that I just WhatsApp, out here we are friends, let what happens in WhatsApp remain in WhatsApp". (Tim, interview, KF 2018)

The statement "what happens in WhatsApp remains in WhatsApp" points to the metaphoric separation of online spaces from real physical spaces and the dangers that abound in the latter spaces. In other words, WhatsApp affords users a disembodied experience.

There were some participants, however, who chose to interact using pseudonyms to escape surveillance. WhatsApp, through its "avatar" feature, enables participants to choose "who they want to be". One can argue that the avatar is a metaphor for identities, including pseudo and "non-identities" which affords the ordinary citizen, especially, a mode of navigating the WhatsApp space without fear of victimization. The excerpt below indicates the reliance of anonymity for safety by some participants within the WhatsApp community network:

> "The good thing with WhatsApp is that you can choose how you want to be known to others you are interacting with. For instance, what you see on my profile as my name (Wizzyboy) is my nickname, and the picture is a cartoon. You see politicians and their supporters are prone to victimizing people if you tell them the truth. So, this way, they cannot tell who I am. I used to go by the same nickname on Facebook too until they changed the system. Now Facebook insists you have to put a real name". (Wizzyboy, interview, EADF 2018)

### 4.2.2. Close community Ties and Friendships

Given the enrolment method, most members are known to one another, further building trust and a sense of safety even though other studies argue that anonymity on internet-based communities cultivates a feeling of safety, increasing participation (Van Heijningen and Van Clief 2017; Matei 2005). The interview excerpt below demonstrates this:

> "When I am in this group, I feel very safe. Most of the people here are my schoolmates and friends. I am sure I can say whatever I want to say no matter how controversial it may be. Nobody can report me to the authority". (Lincy, interview, KF 2018)

For some government officials, community ties are more important than the positions they hold which constructs them as a hindrance to political participation. For instance, during an interview with a policeman, a member of Kabula Forward, he stated that when he and other leaders are in the online community, they are at home as sons and daughters of Kabula to contribute toward the development of their wards, but not to serve the government. A chief in the EADF WhatsApp community, however, indicated that he takes it upon himself to inform group members of new government policies and directives and also caution members to refrain from spreading propaganda in the group or interacting with

others in a manner that could lead to violence, a testament to surveillance in the WhatsApp communities.

### 4.2.3. Banishing "Truant" Members

From participant observations, it was also evident that removing members considered truant is also another strategy to maintain a sense of safety. If one is cautioned not to go against the agreed rules and did not stop, the administrator removed them either as the sole decision-maker or upon the demands of other members. The removing of a member by an administrator is possible through the "remove participant" feature of WhatsApp technology, which acts in this case not only ontically as a "punishment" but also ontologically as a mode of exclusion which denies the banished member a right to belong and participate, consequently, rendering WhatsApp an unsafe space.

### 4.3. Assessing the Effectiveness of Strategies toward Maximalist Participation

The mobilization of actors to participate is equally determined by the strategies discussed in the finding above and other factors impacting on the use of the technological actor affecting the effectiveness of the strategies. For instance, to use WhatsApp, to be an active participating member of a WhatsApp community, one needs to have internet connectivity which costs money and the excerpt below exemplifies this:

"Where is John, he is too quiet today. Could it be he doesn't have data bundles (inserts laughing emoticons)". (Pat, observation, EADF 2018)

Such statements are a common feature in the two WhatsApp communities. While the statement was made in jest or mockery, for John, one of the most active members in this community not having a "clever and quick comeback", it invokes digital inequality and a broader narrative of capitalism, given the reason presumed to be muting John's voice and, consequently, his participation. Studies show despite proliferation of smart phone usage, Africa is still lagging behind when it comes to internet access (Mare 2017).

Participant observations also indicate that there are certain members whose opinion in the virtual communities always elicits robust participation of the other members while others are totally ignored. They receive few or no comments at all on their postings. If the "ignore" behavior persists, these members retreat into silence to avoid the pain of being ignored, though for some who choose to be silent of their own volition, silence affords them safety not to engage in conversations they deem uncomfortable. In extreme instances, a member may even exit the community if they feel their voice is not valued. For instance, one of the participants who exited the Kabula Forward WhatsApp community stated that:

"I left that WhatsApp community because I felt I was treated unfairly. There are people in the group who feel they are more knowledgeable and therefore better than others. If you try to comment on what they have posted, they quickly shut you down. Their supporters who are mostly people who interact a lot with them will also join him in shutting you up. And when you post something on the forum, they all ignore you. So, if you are not wanted in the group, why stay?". (Jerry, interview, KF 2018)

Members who receive a lot of comments on their postings consequently participate more actively in the virtual communities than those who do not. Another former member of the EADF WhatsApp community, however, cited a different reason for exiting.

"I left because I felt that the group was a waste of time. They were not sticking to the goal of community, that is development. There were so many irrelevant posts just filling my phone's space". (Clara, interview, EADF 2018)

Just as is the case when one is "removed" by an administrator, WhatsApp technology affordances play a role in enabling an unwilling participant to exit the group. When one taps to view a WhatsApp group they are part of on their phone, they are given several options, including "exit the group". The difference is where agency is placed. In the first instance, the action is a controlled practice as it can only be performed by an administrator,

in the second instance, power to "leave" metaphoric for freedom to act including avoiding unsafe spaces, is everyone's domain, including the administrator.

## 5. Discussion

*Safe WhatsApp Space or An Illusion?*

The results show that the construction of a safe space for political participation in these WhatsApp community networks is as a result of both human and technology agency. However, it is clear that the same actions contributing to safety could also render such an online space unsafe. For instance, the use of the "add" convention of WhatsApp by administrators to recruit members means the exclusion of anyone not deemed to meet the criteria of belonging. As such, WhatsApp spaces, though heterogeneous networks, can also be understood as "protected enclaves" in a more dialectic conceptualization as both spaces of withdrawal from potential maltreatment in public spaces and as "bases and training grounds for agitational activities directed towards wider publics" (Fraser 1992, p. 124). However, while the enrolment method keeps away individuals who do not belong by virtue of place of origin, it also means that others who come from the said places but do not know of the existence of the online community or are not connected to already existing members are also excluded, impeding their participation. Furthermore, the technological actor limits membership, and consequently participation, to the number of participants a WhatsApp group can hold.

Trust in WhatsApp affordances such as privacy and the end-to-end encryption convention informs participants' view of WhatsApp as safe for political participation. Indeed, other studies also point to these affordances as major contributing factors to WhatsApp safety. Johns and Cheong (2019) present one such study in which they explore how Malaysian social media users navigate censorship through what they term as the "networked-affect." They found that to escape surveillance, participants moved from "public" social media such as Facebook and Twitter (which they deemed made them vulnerable to state surveillance and the resultant seditious laws) to WhatsApp, Telegram, and Messenger which employ end-to-end encryption. Dencik et al. (2016) and Khazraee and Losey (2016) also present the end-to-end encryption feature of WhatsApp as a technological solution to government intrusion and surveillance. The shareability convention of WhatsApp, however, means that privacy afforded by the technology cannot be assured. Members of WhatsApp communities, as was the case in this study, often take screenshots or export messages from WhatsApp and share them on more public platforms or with the authorities. In Kenya, several WhatsApp administrators accused of sharing fake news or hate messages on the platforms have been arrested.

The statement "what happens in WhatsApp remains in WhatsApp", as quipped by a participant, is not only interesting in the way it draws parallels with the adage "what happens in Vegas, stays in Vegas", it also invokes and fits into the debate on the disembodiment/embodiment of cyberspaces and, consequently, whether these spaces can be considered safe. This participant's construction of WhatsApp technology as affording a total separation of the online realm and the offline realm, as in the case of the policeman interviewed, speaks to disembodiment as a strategy through which participants perceive WhatsApp as a safe space. However, disembodiment discourses cannot hold, as seen by the chief's action to caution other members about the nature of their interaction. This means our corporeal bodies cannot be left behind, as argued by Krämer (2008) and Zhao (2005). What, rather, is the case is what Ajana (2005, p. 29) terms as "pseudo embodiment," where bodily perceptions are essential in the construction of self-identity and subjectivity in the cyberspace or a complete embodiment (Asenbaum 2018). In other words, technology and the corporeal body are synthesized within an assemblage.

Since inclusion of members is undertaken through referrals by friends or people known to them, this means that from the beginning, even though online, one enters a familiar territory abound with offline friendship ties and interactions, negating the discourse of disembodiment. As such, the hierarchies and modes of control which result from our offline

identities cannot be completely eliminated in these online spaces. The chief's actions also point to co-surveillance in the WhatsApp communities and not only vertical surveillance, as the chief, being a son of East Asembo, presents another subject position which makes him an equal to the other community members.

Free speech in the WhatsApp spaces, the crux of democracy and, therefore, contributor to maximalist participation in politics, is seen within the discourses of trust and cohesion which result from the sense of community of "ordinariness" and friendship. This aspect, coupled with the fact that these online communities are spatially imagined, contributes to a sense of community and safety. Pang and Woo (2020) argue that relationship building and social cohesion are possible in WhatsApp communities because of the casual, banal conversations that take place frequently among members. These serve to maintain the rather transient networks (Latour 2005). Additionally, Treré (2015, p. 911) attributes the freedom on the networks, what he refers to as a "digital comfort zone", to the digital backstage WhatsApp provides. This constitutes WhatsApp communities as "safe places where activists could express themselves far from official lights of Facebook wall and pages" (Treré 2015, p. 911, citing a respondent, Ernesto). However, such networks are not devoid of conflict (Latour 1999, 2005) as is demonstrated by some participants either leaving the networks voluntarily or being forced out.

There are other contextual factors not related to technology which contribute to the construction of safety on WhatsApp. For instance, the shift of power from the previously powerful to ordinary persons can be attributed to the translocality factor of the WhatsApp community; its connection to the offline space. Since the general election of 2013, the first election under the then-newly dispensed Constitution of 2010; Kenya operates with two tiers of governments—the national and the devolved county governments. What this means is that previously disenfranchised groups in rural areas are now closer to power and, therefore, able to demystify it. The wards, around which Kabula Forward and East Asembo Development Forum are imagined, are the lowest level of representation within elective politics in Kenya.

This proximity to power and political leaders among rural citizens has sparked great interest and, consequently, participation in politics as they finally have a say on service delivery, making it the new site of power. As Gaventa (2001) argues, "it is the local level which historically has been understood as the key site for democracy building and citizen participation" (p. 10), a situation that was reversed in Kenya. The rural in Kenya has, therefore, become a new site of politics and power, bringing in new actors (Hajer et al. 2003) in the quest for political accountability. The locally based actors in the virtual community are more informed about what is happening in the offline place in terms of politics and development and are, therefore, able to engage robustly and from a point of knowledge in political discussions.

How WhatsApp administrators operate, both as a result of technology and human agency, points to the emergence of new centers of power and leadership as seen in the WhatsApp groups studied by Omanga (2019). This illuminates the double-sidedness of technology, where an affordance, in this case facilitating the formation and running of the WhatsApp communities, can also be a constraint in the sense of governmentality, conducting the conduct of others. While the administrator function affords participants inclusion into the communities with the promise of participation, it at the same time curtails that very freedom by, for instance, "punishing truant members" by removing them from the communities.

Despite the existing and potential threats to safety, WhatsApp spaces cannot be dismissed as an illusion, as what is important is how members construct these spaces and the practices and strategies which support their construction. After all, the idea of a safe space is not only desirable but often contested (Stengel and Weems 2010).

## 6. Conclusions

From the analysis, and in answer to the research question of how WhatsApp technology affordances enable/constrain participants' construction of WhatsApp as a safe space for maximalist participation in political WhatsApp communities, it can be construed that, indeed, a safe space is a matter of construction and negotiations of meaning. The findings show the collaboration of all actors in the network in constructing a safe space, strongly influenced by a sense of community rather than only by technology affordances, with the latter responsible for a false sense of security in certain instances, impeding maximalist participation. For instance, relationships previously formed offline are strengthened by frequent online interaction and the possibility of privacy afforded by the WhatsApp technology. The offline and online interactions are some of the performative actions which maintain the WhatsApp community networks by building trust among participants, thus contributing to feelings of safety and, consequently, moving members closer to maximalist participation. Maximalist participation is, however, far from being achieved in the WhatsApp communities studied due to the actions of both human and non-human actors, which either create or strengthen existing power hierarchies, a testament that the networks are not devoid of conflict. Regardless, most participants considered these WhatsApp spaces safe. It can be construed, therefore, that, indeed, a safe space is a matter of construction and negotiations of meaning.

**Funding:** This research received no external funding.

**Institutional Review Board Statement:** The study was conducted in accordance with the Declaration of Helsinki and approved by the Ethics Committee of University of Witwatersrand (protocol code H17/10/19 date of approval) 20 October 2018.

**Informed Consent Statement:** Informed consent was obtained from all subjects involved in the study.

**Data Availability Statement:** Not applicable.

**Conflicts of Interest:** The author declares no conflict of interest.

## References

### *Primary Sources*

Interview with administrator, EADF, 2018.
Interview with chief, EADF, 2018.
Interview with Clara, EADF, 2018.
Interview with elected Member of County Assembly (MCA), EADF, 2018.
Interview with Jerry, KF, 2018.
Interview with Kilian, EADF, 2018.
Interview with Lincy, KF, 2018.
Interview with Lori, KF, 2018.
Interview with Onyango, EADF, 2018.
Interview with Pat, EADF, 2018.
Interview with Phoebe, EADF, 2018.
Interview with policeman, KF, 2018.
Interview with Super Administrator, KF, 2018.
Interview with Tim, KF, 2018.
Interview with Wizzyboy, EADF, 2018.
Kerry, FGD, EADF, 2018.
Pat, Observation, EADF, 2018.
Shifwefwe, FGD, 2018.

### *Secondary Sources*

Aarvik, Per S. 2015. Uchaguzi: An Analysis of the Crowdsourced Election Monitoring in Kenya 2013. Master's thesis, The University of Bergen, Bergen, Norway.
Ajana, Btihaj. 2005. Disembodiment and cyberspace: A phenomenological approach. *Electronic Journal of Sociology* 7: 1–10.

Albrechtslund, Anders. 2008. Online social networking as participatory surveillance. *First Monday* 13: 3. [CrossRef]

Andrejevic, Mark. 2013. Estranged free labor. In *Digital Labor: The Internet as Playground and Factory*. Edited by Trebor Scholz. New York and London: Routledge, pp. 149–64.

Arun, Chinmayi. 2019. On WhatsApp, Rumours, and Lynchings. *Economic and Political Weekly* 54: 30–35.

Asenbaum, Hans. 2018. Cyborg activism: Exploring the reconfigurations of democratic subjectivity in Anonymous. *New Media and Society* 20: 1543–63. [CrossRef]

Bailard, Catie Snow, and Steven Livingston. 2014. Crowdsourcing accountability in a Nigerian election. *Journal of Information Technology and Politics* 11: 349–67. [CrossRef]

Bakhtin, Mikhail. 1981. *The Dialogic Imagination: Four Essays*. Edited by Michael Holquist. Translated by Caryl Emerson, and Michael Holquist. Austin: University of Texas Press, vol. 84, pp. 80–82.

Bauman, Zygmunt, and David Lyon. 2012. *Liquid Surveillance*. Cambridge: Polity Press.

Bennett, Jane. 2004. The force of things: Steps toward an ecology of matter. *Political Theory* 32: 347–72. [CrossRef]

Boyd, Danah. 2010. Social network sites as networked publics: Affordances, dynamics, and implications. In *A Networked Self*. London: Routledge, pp. 47–66.

Bruns, Axel, Jason A. Wilson, and Barry J. Saunders. 2008. Building spaces for hyperlocal citizen journalism. Paper presented at Association of Internet Researchers: Internet Research 9.0: Rethinking Community, Rethinking Place, Copenhagen, Denmark, October 15–18.

Carpentier, Nico. 2007. Translocalism, community media and city. *Culture* 28: 1–32.

Carpentier, Nico. 2009. *Digital Storytelling in Belgium*. Hoboken: Wiley Online Library.

Carpentier, Nico. 2012. The concept of participation. If they have access and interact, do they really participate? *Fronteiras-Estudos Midiáticos* 14: 164–77. [CrossRef]

Carpentier, Nico. 2014. 'Fuck the clowns from Grease!!' Fantasies of participation and agency in the YouTube comments on a Cypriot Problem documentary. *Information, Communication and Society* 17: 1001–16. [CrossRef]

Carpentier, Nico, and Bart Cammaerts. 2006. Hegemony, democracy, agonism and journalism: An interview with Chantal Mouffe. *Journalism Studies* 7: 964–75. [CrossRef]

Carpentier, Nico, Rico Lie, and Jan Servaes. 2003. Community media: Muting the democratic media discourse? *Continuum* 17: 51–68. [CrossRef]

Carpentier, Nico, Peter Dahlgren, and Francesca Pasquali. 2013. The democratic (media) revolution: A parallel genealogy of political and media participation. In *Audience Transformations*. London: Routledge, pp. 131–49.

Clark-Parsons, Rosemary. 2018. Building a digital Girl Army: The cultivation of feminist safe spaces online. *New Media and Society* 20: 2125–44. [CrossRef]

Colom, Anna. 2022. WhatsApp Affordances through an Intersectional Lens: Constructing and Rehearsing Citizenship in Western Kenya. In *International Conference on Social Implications of Computers in Developing Countries*. Cham: Springer, pp. 566–80.

Crawford, Kate. 2009. Following you: Disciplines of listening in social media. *Continuum* 23: 525–35. [CrossRef]

Davis, Mark. 2021. The online anti-public sphere. *European Journal of Cultural Studies* 24: 143–59. [CrossRef]

Dean, Jodi. 2012. *The Communist Horizon*. London: Verso.

Deleuze, Gilles, and Félix Guattari. 1987. *A Thousand Plateaus. Capitalism and Schizophrenia*. Minnesota: University of Minnesota Press.

Dencik, Lina, Arne Hintz, and Jonathan Cable. 2016. Towards data justice? The ambiguity of anti-surveillance resistance in political activism. *Big Data and Society* 3: 2053951716679678. Available online: https://journals.sagepub.com/doi/10.1177/2053951716679678 (accessed on 19 December 2020). [CrossRef]

Diepeveen, Stephanie. 2019. The limits of publicity: Facebook and transformations of a public realm in Mombasa, Kenya. *Journal of Eastern African Studies* 13: 158–74. [CrossRef]

Durak, Hatice Yildiz. 2019. Investigation of nomophobia and smartphone addiction predictors among adolescents in Turkey: Demographic variables and academic performance. *The Social Science Journal* 56: 492–517. [CrossRef]

Dylko, Ivan, and Michael McCluskey. 2012. Media effects in an era of rapid technological transformation: A case of user-generated content and political participation. *Communication Theory* 22: 250–78. [CrossRef]

Elias, Norbert. 1970. Processes of state formation and nation building. In *Transactions of the Seventh World Congress of Sociology*. Genebra: International Sociological Association, pp. 274–84.

Farooq, Gowhar. 2018. Politics of Fake News: How WhatsApp became a potent propaganda tool in India. *Media Watch* 9: 106–17. [CrossRef]

Fisher, Mark. 2009. *Capitalist Realism*. Ropley: Zero Books.

Flyvbjerg, Bent. 2001. *Making Social Science Matter: Why Social Inquiry Fails and How It Can Succeed again*. Cambridge: Cambridge University Press.

Foucault, Michel. 1979. *Discipline and Punish: The Birth of the Prison*. New York: Vintage Books.

Foucault, Michel. 1985. *The Use of Pleasure: The History of Sexuality*. New York: Vintage, vol. 2.

Foucault, Michel. 1994. Truth and Juridical Forms. In *Power—Essential Works of Foucault 1954–1984*. Edited by James D. Faubion. London: Penguin Books, vol. 3.

Fraser, Nancy. 1992. Rethinking the Public Sphere: A Contribution to the Critique of Actually Existing Democracy. In *Habermas and the Public Sphere*. Edited by Craig Calhoun. Boston: MIT Press.

Frederiksen, Bodil Folke. 2011. Print, newspapers and audiences in colonial Kenya: African and Indian improvement, protest and connections. *Africa* 81: 155–72. [CrossRef]

Fuchs, Christian. 2011. *Foundation of Critical Media and Information Studies*. London: Routledge.

Garcia, Betty, and Dorothy Van Soest. 1997. Changing perceptions of diversity and oppression: MSW students discuss the effects of a required course. *Journal of Social Work Education* 33: 119–29. [CrossRef]

Gaventa, John. 2001. Participatory Local Governance: Six Propositions for Development. Paper presented at the Ford Foundation, LOGO Program Officers' Retreat, June Program Officers' Retreat, Brighton, UK, June; Falmer: Institute of Development Studies (IDS).

Gibson, Anna. 2019. Free Speech and Safe Spaces: How Moderation Policies Shape Online Discussion Spaces. *Social Media + Society* 5: 2056305119832588. [CrossRef]

Gil de Zúñiga, Homero, Alberto Ardèvol-Abreu, and Andreu Casero-Ripollés. 2021. WhatsApp political discussion, conventional participation and activism: Exploring direct, indirect and generational effects. *Information, Communication and Society* 24: 201–18. [CrossRef]

Golding, Peter, and Graham Murdock. 1996. Culture, communications, and Political economy. In *Mass Media and Society*. Edited by James Curran and Michael Gurevitch. London: Arnold.

Grinyer, Anne. 2007. The ethics of Internet usage in health and personal narratives research. *Social Research Update* 49: 1–4.

Grosser, Benjamin. 2014. What Do Metrics Want? How Quantification Prescribes Social Interaction on Facebook. *Computational Culture* 4. Available online: http://computationalculture.net/article/what-do-metrics-want (accessed on 3 April 2017).

Hajer, Maarten, Maarten A. Hajer, Hendrik Wagenaar, Robery E. Goodin, and Brian Barry, eds. 2003. *Deliberative Policy Analysis: Understanding Governance in the Network Society*. Cambridge: Cambridge University Press.

Hine, Christine. 2008. Virtual ethnography: Modes, varieties, affordances. In *The SAGE Handbook of Online Research Methods*. London: SAGE Publications, Ltd., pp. 257–70.

Holley, Lyne C., and Sue Steiner. 2005. Safe space: Student perspectives on classroom environment. *Journal of Social Work Education* 41: 49–64. [CrossRef]

Howarth, David, and Steven Griggs. 2006. Metaphor, catachresis and equivalence: The rhetoric of freedom to fly in the struggle over aviation policy in the United Kingdom. *Policy and Society* 25: 23–46. [CrossRef]

International Telecommunication Union (ITU). 2020. Measuring Digital Development: Facts and Figures 2022. Available online: https://www.itu.int/en/ITU-D/Statistics/Pages/facts/default.aspx (accessed on 20 January 2023).

Jenkins, Henry. 2006. *Convergence Culture: Where Old and New Media Collide*. New York: New York University Press.

Jha, Anjali, and Norman Krasner. 2008. Tracking Lost and Stolen Mobile Devices Using Location Technologies and Equipment Identifiers. U.S. Patent 7,446,655, November 4.

Johns, Amelia, and Niki Cheong. 2019. Feeling the chill: Bersih 2.0, state censorship and 'networked affect' on Malaysian social media 2012–2018. *Social Media + Society* 5: 2056305118821801. Available online: https://journals.sagepub.com/doi/10.1177/2056305118821801 (accessed on 20 December 2020). [CrossRef]

Katiambo, David. 2017. #SomeonetellCNN: The agonistic relationship between South and North Media memories. *Journal of Media Critiques* 3: 25–39.

Katiambo, David. 2019. Incivility in Social Media as Agonistic Democracy? A Discourse Theory Analysis of Dislocation and Repair in Select Government Texts in Kenya. Doctoral dissertation, University of South Africa, Pretoria, South Africa.

Katiambo, David, and Fred Orina Ochoti. 2021. Illusion in the Digitised Public Sphere: Reading WhatsApp through the Grammar of Phantasmagoria. *Critical Arts* 35: 39–54. [CrossRef]

Katzer, Catarina, Detlef Fetchenhauer, and Frank Belschak. 2009. Cyberbullying: Who are the victims? A comparison of victimization in Internet chatrooms and victimization in school. *Journal of Media Psychology* 21: 25–36. [CrossRef]

Khazraee, Emad, and James Losey. 2016. Evolving repertoires: Digital media use in contentious politics. *Communication and the Public* 1: 39–55. [CrossRef]

Kimari, Wangui, and Jacob Rasmussen. 2010. Setting the agenda for our leaders from under a tree. *Nokoko* 1: 131–59.

Kozinets, Robert V. 2002. The field behind the screen: Using netnography for marketing research in online communities. *Journal of Marketing Research* 39: 61–72. [CrossRef]

Kozinets, Robert V. 2007. 10 Netnography 2.0. In *Handbook of Qualitative Research Methods in Marketing*. Cheltenham: Edward Elgar Publishing, p. 129.

Kozinets, Robert V. 2010. *Netnography: Ethnographic Research in the Age of the Internet*. Thousand Oaks: Sage Publications Ltd.

Kozinets, Robert V. 2015. *Netnography. The International Encyclopedia of Digital Communication and Society*. New York: John Wiley & Sons, pp. 1–8.

Krämer, Sybille. 2008. Does the Body Disappear? A Comment on Computer Generated Spaces. In *Paradoxes of Interactivity. Perspectives for Media Theory, Human-Computer Interaction, and Artistic Investigations*. Edited by Uwe Seifert, Jin Hyun Kim and Anthony Moore. Bielefeld: Transcript, pp. 26–42.

Laclau, Ernesto, and Chantal Mouffe. 1985. *Hegemony and Socialist Strategy: Towards a Radical Democratic Politics*. London and New York: Verso.

Latour, Bruno. 1996. On actor-network theory: A few clarifications. *Soziale Welt* 47: 369–81.

Latour, Bruno. 1999. On recalling ANT. *The Sociological Review* 47: 15–25. [CrossRef]

Latour, Bruno. 2005. *Reassembling the Social—An Introduction to Actor Network Theory*. Oxford: Oxford University Press.

Law, John. 1987. Technology and heterogeneous engineering: The case of Portuguese expansion. In *The Social Construction of Technological Systems: New Directions in the Sociology and History of Technology*. Cambridge: MIT Press, vol. 1, pp. 1–134.

Law, John. 1992. Notes on the theory of the actor-network: Ordering, strategy, and heterogeneity. *Systems Practice* 5: 379–93. [CrossRef]

Law, John. 2009. Actor network theory and material semiotics. *The New Blackwell Companion to Social Theory* 3: 141–58.

Law, John, and Michel Callon. 1988. Engineering and sociology in a military aircraft project: A network analysis of technological change. *Social Problems* 35: 284–97. [CrossRef]

Leary, Mark R., and Robin Mark Kowalski. 1995. *Social Anxiety*. New York: Guilford Press.

Livingston, Steven. 2011. *Africa's Evolving Infosystems: A Pathway to Security and Stability*. Washington, DC: Africa Centre For Strategic Studies, National Defence University.

Livingston, Steven, and Gregor Walter-Drop, eds. 2014. *Bits and Atoms: Information and Communication Technology in Areas of Limited Statehood*. Oxford: Oxford University Press.

Livingstone, Sonia, Leslie Haddon, Anke Görzig, and Kjartan Ólafsson. 2011. *Risks and Safety on the Internet: The Perspective of European Children: Full Findings and Policy Implications from the EU Kids Online Survey of 9–16 Year Olds and Their Parents in 25 Countries*. London: EU Kids Online Network.

Livingstone, Sonia, Kjartan Ólafsson, Ellen J. Helsper, Francisco Lupiáñez-Villanueva, Giuseppe A. Veltri, and Frans Folkvord. 2017. Maximizing opportunities and minimizing risks for children online: The role of digital skills in emerging strategies of parental mediation. *Journal of Communication* 67: 82–105. [CrossRef]

Lovink, Geert. 2011. *Networks without a Cause*. Amsterdam: Institute of Network Cultures.

Lucero, Leanna. 2017. Safe spaces in online places: Social media and LGBTQ youth. *Multicultural Education Review* 9: 117–28. [CrossRef]

Lyon, David. 1994. *The Electronic Eye: The Rise of Surveillance Society*. Cambridge: Polity Press.

Mahlouly, Dounia. 2013. Rethinking the public sphere in a digital environment: Similarities between the eighteenth and the twenty-first centuries. *ESharp* 20: 1–21.

Mann, Steve, Jason Nolan, and Barry Wellman. 2003. Sousveillance: Inventing and Using Wearable Computing Devices for Data Collection in Surveillance Environments. *Surveillance and Society* 1: 331–55. [CrossRef]

Mare, Admire. 2017. Tracing and archiving 'constructed' data on Facebook pages and groups: Reflections on fieldwork among young activists in Zimbabwe and South Africa. *Qualitative Research* 17: 645–63. [CrossRef]

Marshall, Martin N. 1996. Sampling for qualitative research. *Family Practice* 13: 522–26. [CrossRef]

Matei, Sorin Adam. 2005. From counterculture to cyberculture: Virtual community discourse and the dilemma of modernity. *Journal of Computer-Mediated Communication* 10: JCMC1031. [CrossRef]

Mbeke, Oriare Peter. 2008. *Background Note: The Media, Legal, Regulatory and Policy Environment in Kenya*. A Historical Briefing. Nairobi: Commissioned by the BBC World Service Trust.

Mintzberg, Henry, Ken G. Smith, and Michael A. Hitt, eds. 2005. Great minds in management: The process of theory development. In *Developing Theory about the Development of Theory*. Cheltenham: Edward Elgar Publishing, pp. 355–72.

Mouffe, Chantal. 2005. *On the Political*. London: Routledge.

Mukhongo, Lynette Lusike. 2020. Participatory Media Cultures: Virality, Humour, and Online Political Contestations in Kenya. *Africa Spectrum* 55: 148–69. [CrossRef]

Munene, Jane Wamaitha, and D. Reckson Thakhathi. 2017. An analysis of capacities of civil society organizations (CSOs) involved in promotion of community participation in governance in Kenya. *Journal of Public Affairs* 17: e1668. [CrossRef]

Murthy, Dhiraj. 2008. Digital ethnography: An examination of the use of new technologies for social research. *Sociology* 42: 837–55. [CrossRef]

Mutahi, Patrick, and Brian Kimari. 2017. *The Impact of Social Media and Digital Technology on Electoral Violence in Kenya*. Falmer: Institute of Development Studies (IDS).

Ogola, George. 2011. The political economy of the media in Kenya: From Kenyatta's nation-building press to Kibaki's local-language FM radio. *Africa Today* 57: 77–95. [CrossRef]

Ogola, George. 2015. Social media as a heteroglossic discursive space and Kenya's emergent alternative/citizen experiment. *African Journalism Studies* 36: 66–81. [CrossRef]

Okoth, Grace Brenda W. 2020. How Kenyans on Twitter use visuals as a form of political protest. *Journal Kommunikation. Medien*, 1–27.

Omanga, Duncan. 2019. WhatsApp as 'digital publics': The Nakuru Analysts and the evolution of participation in county governance in Kenya. *Journal of Eastern African Studies* 13: 175–91. [CrossRef]

Onwuzurike, Lucky, and Emiliano De Cristofaro. 2016. Experimental Analysis of Popular Smartphone Apps Offering Anonymity, Ephemerality, and End-to-End Encryption. *arXiv* arXiv:1510.04083. [CrossRef]

Pang, Natalie, and Yue Ting Woo. 2020. What about WhatsApp? A systematic review of WhatsApp and its role in civic and political engagement. *First Monday* 25: 12. [CrossRef]

Papacharissi, Zizi. 2009. The virtual geographies of social networks: A comparative analysis of Facebook, LinkedIn and ASmallWorld. *New Media and Society* 11: 199–220. [CrossRef]

Pearce, Katie E., and Jessica Vital. 2015. Performing Honor Online: The Affordances of Social Media for Surveillance and Impression Management in an Honor Culture. *New Media and Society* 18: 2595–612. [CrossRef]

Posey, Clay, Paul Benjamin Lowry, Tom L. Roberts, and Timothy Selwyn Ellis. 2010. Proposing the online community self-disclosure model: The case of working professionals in France and the UK who use online communities. *European Journal of Information Systems* 19: 181–95. [CrossRef]

Postill, John, and Sarah Pink. 2012. Social media ethnography: The digital researcher in a messy web. *Media International Australia* 145: 123–34. [CrossRef]

Reed, Michael. 2005. Reflections on the 'realist turn'in organization and management studies. *Journal of Management Studies* 42: 1621–44. [CrossRef]

Republic of Kenya. 2018. *Computer Misuse and Cybercrimes Act No. 5 of 2018.* Nairobi: Government Printers.

Rheingold, Howard. 2000. *The Virtual Community: Homesteading on the Electronic Frontier*. Cambridge: MIT Press.

Ringmar, Erik. 2018. The problem with performativity: Comments on the contributions. *Journal of International Relations and Development* 22: 899–908. [CrossRef]

Roestone Collective. 2014. Safe space: Towards a reconceptualization. *Antipode* 46: 1346–65. [CrossRef]

Roy, Kevin, Anisa Zvonkovic, Abbie Goldberg, Elizabeth Sharp, and Ralph LaRossa. 2015. Sampling richness and qualitative integrity: Challenges for research with families. *Journal of Marriage and Family* 77: 243–60. [CrossRef]

Sassetti, Francisca. 2019. Social Media and crowdsourced election monitoring: Prospects for election transparency in Sub-Saharan Africa. *Politikon: The IAPSS Journal of Political Science* 42: 7–39. [CrossRef]

Scheuerman, Morgan Klaus, Stacy M. Branham, and Foad Hamidi. 2018. Safe spaces and safe places: Unpacking technology-mediated experiences of safety and harm with transgender people. *Proceedings of the ACM on Human-Computer Interaction* 2: 1–27. [CrossRef]

Schroer, Markus. 2019. Spatial Theories/Social Construction of Spaces. In *The Wiley Blackwell Encyclopedia of Urban and Regional Studies*. Hoboken: John Wiley & Sons Press, pp. 1–11.

Seabrook, John. 1998. *Deeper: My Two-Year Odyssey in Cyberspace*. New York: Simon and Schuster Trade.

Shilaho, Westen Kwatemba. 2018. The Kenyan State and the Ethnicity Challenge. In *Political Power and Tribalism in Kenya*. Cham: Palgrave Macmillan, pp. 29–49.

Shirky, Clay. 2009. *Here Comes Everybody: How Change Happens When People Come Together*. London: Penguin.

Skågeby, Jörgen. 2011. Online ethnographic methods: Towards a qualitative understanding of virtual community practices. In *Handbook of Research on Methods and Techniques for Studying Virtual Communities: Paradigms and Phenomena*. Hershey: IGI Global, pp. 410–28.

Soetan, Funmi. 2012. Ory Okolloh: Kenyan Pundit, Google Policy Manager for Africa. Ventures Africa. Available online: https://venturesafrica.com/ory-okolloh-kenyan-pundit-google-policy-manager-for-africa/ (accessed on 10 February 2021).

Srinivasan, Sharath, Stephanie Diepeveen, and George Karekwaivanane. 2019. Rethinking publics in Africa in a digital age. *Journal of Eastern African Studies* 13: 2–17. [CrossRef]

Stake, Robert E. 1995. *The Art of Case Study Research*. Thousand Oaks: SAGE.

Stengel, Barbara S., and Lisa Weems. 2010. Questioning safe space: An introduction. *Studies in Philosophy and Education* 29: 505–7. [CrossRef]

Sugiura, Lisa, Rosemary Wiles, and Catherine Pope. 2017. Ethical challenges in online research: Public/private perceptions. *Research Ethics* 13: 184–99. [CrossRef]

Svensson, Jakob. 2012. Social media and the disciplining of visibility: Activist participation and relations of power in network societies. *European Journal of ePractice* 16: 16–28.

Treré, Emiliano. 2015. Reclaiming, proclaiming, and maintaining collective identity in the #YoSoy132 movement in Mexico: An examination of digital frontstage and backstage activism through social media and instant messaging platforms. *Information, Communication and Society* 18: 901–15.

Treré, Emiliano. 2018. The sublime of digital activism: Hybrid media ecologies and the new grammar of protest. *Journalism and Communication Monographs* 20: 137–48. [CrossRef]

Tuikka, Anne Marie, Chau Nguyen, and Kai K. Kimppa. 2017. Ethical questions related to using netnography as research method. *ORBIT Journal* 1: 1–11.

Van Heijningen, Maaike, and Lindsay Van Clief. 2017. Enabling Online Safe Spaces: A Case Study of Love Matters Kenya. *IDS Bulletin* 48: 1. [CrossRef]

Vasudeva, Feeza, and Nicholas Barkdull. 2020. WhatsApp in India? A case study of social media related lynchings. *Social Identities* 26: 574–89. [CrossRef]

Velasquez, Alcides, Andrea M. Quenette, and Hernando Rojas. 2021. WhatsApp political expression and political participation: The role of ethnic minorities' group solidarity and political talk ethnic heterogeneity. *International Journal of Communication* 15: 22.

Wanyande, Peter. 2009. *Civil Society and Transition Politics in Kenya: Historical and Contemporary Perspectives*. Nairobi: African Research and Resource Forum (ARRF).

Wasserman, Herman. 2011. Mobile phones, popular media, and everyday African democracy: Transmissions and transgressions. *Popular Communication* 9: 146–58. [CrossRef]

Wenger, Etienne. 1998. *Communities of Practice: Learning, Meaning, and Identity*. Cambridge: Cambridge University Press.

Willems, Wendy. 2015. Mediation, power and civic agency in Africa. In *The Routledge Companion to Alternative and Community Media*. Edited by Chris Atton. London: Routledge, pp. 88–89.

Willems, Wendy, and Winston Mano, eds. 2017. *From Audiences to Users: Everyday Media Culture in Africa*. New York: Routledge.

Xun, Jiyao, and Jonathan Reynolds. 2010. Applying netnography to market research: The case of the online forum. *Journal of Targeting, Measurement and Analysis for Marketing* 18: 17–31. [CrossRef]

Yin, Robert K. 2009. *Case Study Research: Design and Methods*. Thousand Oaks: SAGE, vol. 5.

Zanchetta, Chiara, Hannah Schiff, Carolina Novo, Sandra Cruz, and Carlos Vaz de Carvalho. 2022. Generational Inclusion: Getting Older Adults Ready to Own Safe Online Identities. *Education Sciences* 12: 715. [CrossRef]

Zhao, Shanyang. 2005. The digital self: Through the looking glass of telecopresent others. *Symbolic Interaction* 28: 387–405. [CrossRef]

