# Peer review of "In Pursuit of a “Safe” Space for Political Participation: A Study of Selected WhatsApp Communities in Kenya"

_journalmedia, doi:10.3390/journalmedia4020032_

Round 1
Reviewer 1 Report
The text analyzes an interesting topic, the way in which private interpersonal social networks (whatsapp) are used for political communication. However, some observations about the research can be made.
Some of the data on mobile phone use or Internet consume in Africa is very old (v.g. Wasserman 2011; Etzo & Collender 2010; Hahn & Kibora 2008). They should be replaced by the most recent data available, as the subject requires it.
The two communities analyzed are very small (KF has 230 members, EADF has 182 members). In the methodology there are no references to its representativeness or to the number of similar communities that exist. East Asembo and Kabula are two Wards in Kenya, of which basic information should be provided so that readers can understand the scope of the study, as they may have specific themes, etc.
Due to the way in which WhatsApp communities are created (at the invitation of the administrators), the possible biases that the sample could have should be considered. It would also be useful to provide some basic information about the people interviewed, at least for guidance: age, gender, area in which they work, etc. With less relevance, the statements of the people interviewed should be enclosed in quotation marks.
The study focuses on the way in which communities are managed, and how they are intended to be spaces for free participation. The basis for this is the interview with 17 people who belong or had belonged to these networks. Though, it is not very clear how the content analysis of the messages published during that period is used in the investigation.
There is also no mention of the topics that are being discussed, and the way in which safe spaces are needed for those debates. If the study focuses on the way in which the communities themselves are managed and not so much on their themes, it should focus on the way in which users are managed, how often they participate, etc. All of this could make it possible to characterize at least these communities.
In short, the text is interesting because of the subject it deals with, which certainly deserves to be investigated. On the contrary, it has some relevant gaps: there is no information that allows interpreting the representativeness of the selected sample; The number of messages or the activity of said networks are not specified either; the data is quite old, in terms of social networks (5 years); the conclusions are very generic, and the link with the specific issues (political, local) of these regions (East Asembo and Kabula) is not worked on.
In order to focus on the way in which safe communities are generated for political debate, more information should be provided on the way in which such communities are managed, etc.
Author Response
Dear Reviewer,
Thank you so much for the useful recommendations. I have responded to them as follows:
Reviewer: Some of the data on mobile phone use or Internet consume in Africa is very old (v.g. Wasserman 2011; Etzo & Collender 2010; Hahn & Kibora 2008). They should be replaced by the most recent data available, as the subject requires it.
Author: The references have been replaced with recent ones.
Reviewer: The two communities analyzed are very small (KF has 230 members, EADF has 182 members). In the methodology there are no references to its representativeness or to the number of similar communities that exist. East Asembo and Kabula are two Wards in Kenya, of which basic information should be provided so that readers can understand the scope of the study, as they may have specific themes, etc.
Author: More information has been provided towards this. The cases were instrumental, the author was not thriving for representativeness but rather unique characteristics of the cases, that is, give the goal of the communities.
With regards to numbers, at the time of the study, WhatsApp groups had a limit of 256. Moreover, the numbers kept changing due to more additions or members leaving the communities. Data was also deliberately collected at that particular time after an election, to avoid biases in composition because during the months towards an election, the communities were full, but those who join just for the elections left the groups afterwards.
Reviewer: Due to the way in which WhatsApp communities are created (at the invitation of the administrators), the possible biases that the sample could have should be considered. It would also be useful to provide some basic information about the people interviewed, at least for guidance: age, gender, area in which they work, etc. With less relevance, the statements of the people interviewed should be enclosed in quotation marks.
Author: More information on respondents has been provided and statements enclosed in quotation marks
Reviewer: The study focuses on the way in which communities are managed, and how they are intended to be spaces for free participation. The basis for this is the interview with 17 people who belong or had belonged to these networks. Though, it is not very clear how the content analysis of the messages published during that period is used in the investigation.
Author: There was an omission of the word “each” in the sample. The 5 were per community. So instead of using the word each (it was not fittig well grammatically) I simply put the numbers in total.
There is also no mention of the topics that are being discussed, and the way in which safe spaces are needed for those debates. If the study focuses on the way in which the communities themselves are managed and not so much on their themes, it should focus on the way in which users are managed, how often they participate, etc. All of this could make it possible to characterize at least these communities.
Author: How these communities are managed was part of what was investigated in this study. However, it formed the basis of another chapter. Still, I have discussed the role of the administrators in relation to the theme of safety in paper. I have also included questions the participants were responding to to point to the themes being discussed in the paper.
Reviewer 2 Report
This is an interesting paper and has a potential to contribute to networked public sphere, political participation and digital activism. The manuscript makes a substantial input into the body of scholarship in this area but will need further refining.
Abstract
Adequate. However, the last sentence in the abstract is unclear and will need to be revised.
Introduction
Referencing error on page 1, line 26. Also, paragraph 1 of the introduction should be revised. It is unclear as it stands. Also, paragraph 2 lines 35 and 36, the author(s) did not close the quotation. The author(s) should look at and revise page 1, line 44: “Hence WhatsApp in and of itself was not responsible for the mob violence, vigilantism, and collapse of the rule of law”. It is unclear what the author(s) are trying to say here.
The author(s) should also watch out for the use of the term “Chapter” to describe this paper. See page 2, line 92 and page 3, line 98.
Literature: This needs to be reorganised and revised substantially. The literature review will need to be improved in two areas: Review of ANT and proper review of Applications such as Ushahidi. First, the author(s) mentioned how important Actor Network Theory (ANT) is to their study. It was mentioned to be significant to the study in the abstract, introduction, methodology, findings and discussion. This illustrates that ANT is seen to be crucial to the study. The issue is that this vital aspect of this paper was not reviewed properly. I suggest that the author(s) conduct a robust and in-depth literature review around ANT to enable the reader follow the theory in order parts of the paper.
Second, the author(s) should cast their net a bit wide to include studies in other parts of Africa to enable them conduct an updated form of literature review. For example, in terms of election monitoring via Ushahidi, the author(s) mentioned it but could have conducted further review here. In page 4, line 176, you will need to study the works for Livingston and his colleagues to improve this review. See the following (Bailard and Livingston, 2014; Siegle and Livingston, 2013; Sassetti, 2019; Livingston, 2011; Livingstone and Water-Drop, 2014).
Methods: The method adopted for this study is adequate. However, what was the nature of the researchers’ participation in the two groups. Are they members who researched the group without the knowledge of the group members or are they just researchers and were known as such by the group members? If the group members were aware, what type of limitation can be drawn from the data and how does that impact the findings of the study? If the group members were not aware, what are the ethical issues that such issue raises and how did the researchers mitigate the ethical issue.
Findings and Discussion
The finding or result is adequate. However, I would like to hear the voice of the interview participants in greater depth than delineated, particularly around those that either left the group or participated less. Also, once the author(s) have reviewed ANT, they will need to use the references from the ANT review to buttress and support their result and discussion. Based on all these, I feel that the manuscript will need to under major changes before it could be ready for publication.
I would like to thank the editor for giving me the opportunity to review this potentially excellent manuscript about WhatsApp and safe space.
Reference
Bailard, C. S., & Livingston, S. (2014). Crowdsourcing accountability in a Nigerian election. Journal of Information Technology & Politics, 11(4), 349-367.
Livingston, S., & Walter-Drop, G. (Eds.). (2014). Bits and atoms: Information and communication technology in areas of limited statehood. Oxford University Press.
Livingston, S. (2011). Africa's evolving infosystems: A pathway to security and stability. NATIONAL DEFENSE UNIV WASHINGTON DC AFRICA CENTER FOR STRATEGIC STUDIES.
Sassetti, F. (2019). Social Media and crowdsourced election monitoring: prospects for election transparency in Sub-Saharan Africa. Politikon: The IAPSS Journal of Political Science, 42, 7-39.
Siegle, J., Livingston, S., & Walter-Drop, G. (2014). ICT and accountability in areas of limited statehood. Bits and Atoms: Information and Communication Technology in Areas of Limited Statehood, 61.
Author Response
Abstract
Reviewer: Adequate. However, the last sentence in the abstract is unclear and will need to be revised.
Author: Revised
Introduction
Reviewer: Referencing error on page 1, line 26. Also, paragraph 1 of the introduction should be revised. It is unclear as it stands. Also, paragraph 2 lines 35 and 36, the author(s) did not close the quotation. The author(s) should look at and revise page 1, line 44: “Hence WhatsApp in and of itself was not responsible for the mob violence, vigilantism, and collapse of the rule of law”. It is unclear what the author(s) are trying to say here.
Author: Corrected
The author(s) should also watch out for the use of the term “Chapter” to describe this paper. See page 2, line 92 and page 3, line 98.
Author: Corrected
Reviewer: Literature: This needs to be reorganised and revised substantially. The literature review will need to be improved in two areas: Review of ANT and proper review of Applications such as Ushahidi. First, the author(s) mentioned how important Actor Network Theory (ANT) is to their study. It was mentioned to be significant to the study in the abstract, introduction, methodology, findings and discussion. This illustrates that ANT is seen to be crucial to the study. The issue is that this vital aspect of this paper was not reviewed properly. I suggest that the author(s) conduct a robust and in-depth literature review around ANT to enable the reader follow the theory in order parts of the paper.
Author: Reviewed ANT (introduced a theory review section)
Second, the author(s) should cast their net a bit wide to include studies in other parts of Africa to enable them conduct an updated form of literature review. For example, in terms of election monitoring via Ushahidi, the author(s) mentioned it but could have conducted further review here. In page 4, line 176, you will need to study the works for Livingston and his colleagues to improve this review. See the following (Bailard and Livingston, 2014; Siegle and Livingston, 2013; Sassetti, 2019; Livingston, 2011; Livingstone and Water-Drop, 2014).
Author: Suggested articles reviewed and included
Reviewer: Methods: The method adopted for this study is adequate. However, what was the nature of the researchers’ participation in the two groups. Are they members who researched the group without the knowledge of the group members or are they just researchers and were known as such by the group members? If the group members were aware, what type of limitation can be drawn from the data and how does that impact the findings of the study? If the group members were not aware, what are the ethical issues that such issue raises and how did the researchers mitigate the ethical issue.
Author: A section on ethical considerations introduced and issues raised discussed.
Findings and Discussion
The finding or result is adequate. However, I would like to hear the voice of the interview participants in greater depth than delineated, particularly around those that either left the group or participated less. Also, once the author(s) have reviewed ANT, they will need to use the references from the ANT review to buttress and support their result and discussion. Based on all these, I feel that the manuscript will need to under major changes before it could be ready for publication.
Author: Recommendation followed.
Round 2
Reviewer 1 Report
The reviewer wishes to thank the authors for the responses and explanations they have given.
Some of the issues have already been clarified, but some of the concerns remain.
There is still a profound lack of context for the study. The reasons for choosing these two communities in particular should be specified, and say in what way they are representative of a global reality. If they are exceptional, also why. Case studies always require a context to assess their relevance.
The information about the people interviewed is clearer, although it would be good to identify the gender, the age (even if it is the range), if possible. In addition to the theoretical methodological explanations, which are highly appreciated, it would also be necessary to describe the specific way in which the content analysis was carried out.
If there is complementary data on the same research in relation to the themes, etc., even if they are published elsewhere, they should be cited at least synthetically. They are relevant to understand the work of these communities.
If the study focuses on the way in which the communities themselves are managed, the conclusions and discussion should try to be more analytical about the way in which this particular work illustrates the global problem of participation and moderation in Kenya or this type of groups.
Thank you very much for your answers.
Author Response
Thank you for reviewing my article. I respond as follows:
Reviewer: There is still a profound lack of context for the study. The reasons for choosing these two communities in particular should be specified and say in what way they are representative of a global reality. If they are exceptional, also why. Case studies always require a context to assess their relevance.
Response: The article increasingly builds context of the study from the abstract, introduction and literature review sections (particularly, section 1.2 “The elusive safe space for political participation in Kenya” to the methodology sections I also incorporate context in presentation of findings. On why the two communities were chosen, I have provided an explanation in the methodology section 3.1 ‘the cases”.
Reviewer: The information about the people interviewed is clearer, although it would be good to identify the gender, the age (even if it is the range), if possible.
Response: Has been taken care of in lines 418-419.
Reviewer: In addition to the theoretical methodological explanations, which are highly appreciated, it would also be necessary to describe the specific way in which the content analysis was carried out.
Response: I have described how analysis was carried out in section 3.3 “Procedure for data analysis”.
Reviewer: If there is complementary data on the same research in relation to the themes, etc., even if they are published elsewhere, they should be cited at least synthetically. They are relevant to understand the work of these communities.
Response: The themes have been built up from the findings to the discussion section. In the discussion section, I discuss the emerging sub themes including trust in both human and technological actor, sense of community, disembodiment/embodiment, proximity to power. These sub themes are instrument in explaining the major theme, being that a safe space is a matter of construction. Therefore, despite the many challenges, participants are of the overall view that WhatsApp is a safe place for political participation.
Reviewer: If the study focuses on the way in which the communities themselves are managed, the conclusions and discussion should try to be more analytical about the way in which this particular work illustrates the global problem of participation and moderation in Kenya or this type of groups.
Response: the main focus of the article is how various discourses in the communities resulting from the interaction of human and technological actor contribute to the construction of WhatsApp as a safe(unsafe) space for political participation. How the communities are managed are one of the many discourses which emerged. In addition to the information already provided in the findings on this, for instance; rules of engagement, joining or leaving the communities, “punishing truant members”. I have added a discussion on this in lines 942-949.
I hope I have satisfactorily addressed the concerns.
Reviewer 2 Report
This is an interesting paper contribute to networked public sphere, political participation and digital activism. The manuscript makes a substantial input into the body of scholarship in this area. This reworked version have substantially addressed my concerns. I am confident with the paper and recommend publication.
I would like to thank the editor for giving me the opportunity to review this excellent manuscript. I wish the author(s) best of luck.
Author Response
Thank you for your review.
Round 3
Reviewer 1 Report
The reviewer would like to than the author(s) for the improvements in the manuscript.
Some minor suggestions may include a bit more precise description of the participants in the sample and an reference in the discussion to prior publications on the topic by the authors about the same topic.
Best of luck with further research and thank you very much for the updates.